# Analysis of the Association between Galectin-3 Concentration in Tears and the Severity of Dry Eye Disease: A Case-Control Study

**DOI:** 10.3390/jcm11010066

**Published:** 2021-12-23

**Authors:** Miki Hata-Mizuno, Yuichi Uchino, Miki Uchino, Shigeto Shimmura, Yoko Ogawa, Kazuo Tsubota, Kazuno Negishi

**Affiliations:** Department of Ophthalmology, Keio University School of Medicine, Tokyo 160-8582, Japan; morethan3miles@gmail.com (M.H.-M.); uchinomiki@yahoo.co.jp (M.U.); shige@z8.keio.jp (S.S.); yoko@z7.keio.jp (Y.O.); tsubota@z3.keio.jp (K.T.); kazunonegishi@keio.jp (K.N.)

**Keywords:** galectin-3, galectin-3C, dry eye disease, dry eye syndrome, keratoconjunctivitis sicca, keratitis sicca

## Abstract

This study aimed to investigate the relationship between the severity of dry eye disease (DED) and galectin-3 concentration (gal-3) and its cleavage (gal-3C) in tear fluid. Twenty-eight DED patients and 14 controls were recruited at Keio University Hospital. The lissamine green conjunctival staining (LG) score, fluorescein corneal staining (FL) score, tear film break-up time (TBUT), Schirmer’s test, and ocular symptoms questionnaire score (dry eye questionnaire score, DEQS) were evaluated. Furthermore, the correlation between these parameters and the concentrations of gal-3 in tears (ng/µg) and the detection rate of gal-3C (%) were analyzed. Gal-3 concentration in tears was positively correlated with the LG score (R = 0.60, *p* < 0.01), FL score (R = 0.49, *p* < 0.01), and DEQS (R = 0.45, *p* < 0.01), and negatively correlated with the TBUT score (R = −0.40, *p* < 0.01) and Schirmer’s I value (R = −0.36, *p* < 0.01). The detection rate of gal-3C in tears was significantly associated with the severity of DED, especially with the LG (*p* < 0.01) and FL (*p* < 0.01) scores. Therefore, the concentration of gal-3 and the detection rate of gal-3C in tears had a significant relationship with the severity of ocular surface barrier disruption.

## 1. Introduction

Dry eye disease (DED) is a common ocular disorder defined as a multifactorial disease of tears and the ocular surface, which may result in symptoms of discomfort and visual disturbance [1]. A previous work showed that DED is now a social problem due to symptoms that reduce the quality of life and work productivity of patients [2].

The tear film consists of an oily, an aqueous, and a mucin layer [3,4,5]. Mucins, large high-molecular-weight glycosylated glycoproteins, are classified into two types: gel-forming mucins secreted by the conjunctival goblet cells and transmembrane mucins (MUCs) expressed by the stratified corneal and conjunctival epithelium, detected in the aqueous and mucin layers, respectively [6,7,8,9]. MUCs, which are composed of the core protein and glycochains, are expressed on the microplicae tips of the epithelial cells and extend up to 500 nm into the tear film [7,10,11,12]. They help protect cells from pathogens and mechanical and chemical damage [13,14]. The mucin layer covering the ocular surface lowers surface tension and improves tear wettability [15,16,17]. MUC1, MUC4, and MUC16 [18,19,20] have been mainly detected in the stratified corneal and conjunctival epithelium. The molecular weight of MUC16 is the largest compared to that of other mucins; therefore, it predominantly covers the ocular surface and makes a substantial contribution to the ocular surface barrier function [21]. Galectin-3 (gal-3), which is a member of the lectin family, is indispensable for the formation of this glycocalyx barrier formed by MUC16.

Of the 15 mammalian galectins identified [22,23,24], gal-3 is a 35-kD chimeric protein produced by the corneal and conjunctival epithelium and consists of a C-terminal carbohydrate recognition domain (CRD, lectin domain) and the N-terminal domain. The CRD contains 110–130 unusually flexible amino acids that enable the binding of β-galactoside-binding proteins. The N-terminal domain has 7–14 repeats of a nine-amino acid sequence that is essential for gal-3 polymerization [25,26,27]. This is also the terminal sensitive to proteolysis by matrix metalloproteinases such as MMP-2 and MMP-9 [28,29]. Gal-3 undergoes monomerization or dimerization in the absence of binding ligands and multimerization in the presence of carbohydrate-binding ligands. The cleavage product of gal-3 (gal-3C) results in a truncated gal-3 with 22–27 kDa peptides, containing the C-terminal domain, but lacking the N-terminal domain. This increases affinity to glycans and diminishes capacity for self-association. In the absence of its N-terminal domain, gal-3C binds to endothelial cells [30] and laminin [29]. Gal-3 also has a greater affinity for galactose-terminated glycans, such as lactose and N-acetyllactosamine, rather than simple galactose [29].

After gal-3 is secreted into the extracellular space from ocular surface epithelial cells, it combines with the *O*-glycans of the MUCs, especially MUC16, and forms the ocular surface glycocalyx barrier [27,31,32]. This barrier plays an important role as a lubricant [33], moisturizer, barrier [34], and in stimulation of pathways involved in the innate immune response [27,35].

In DED cases, the expression of MUCs are altered [36], resulting in the disruption of mucin-gal-3 binding in the glycocalyx, and considerably allowing easier detection of gal-3 protein in tears than normal. Decreased MUCs affect tear stability on the ocular surface and cause subjective symptoms in patients with DED [15]. Therefore, it is important to treat it qualitatively, although there is no method to clinically detect the disruption of MUCs on the human ocular surface. Therefore, in this study, we focused on gal-3 examination as an alternative evaluation method. Our previous clinical study revealed that gal-3 and gal-3C could be detected better in tears of patients with DED compared to normal individuals, as gal-3 was released from MUC16, the expression of which decreased on the ocular surface [37]. However, there have been no reports on the relationship between the severity of DED and disruption of the glycocalyx barrier. Herein, we investigated the correlation between the disruption of the glycocalyx barrier and the general ophthalmic examination findings in patients with DED. Furthermore, we also studied the association between the severity of DED and the detection rate of gal-3C.

## 2. Materials and Methods

### 2.1. Study Design and Population

The study population included participants who visited the clinic at Keio University from October 2017 to September 2019. We applied the Japanese DED diagnostic criteria for enrolling DED patients in this study. None of the healthy controls had any eye symptoms or eye diseases. The exclusion criteria for both groups included inflammatory diseases, such as ocular allergies, use of contact lens, history of eye surgery within the previous year, and smoking history, determined by an institutional review board-approved questionnaire. 

### 2.2. Diagnosis of Dry Eye Disease (DED)

Briefly, Japanese diagnostic criteria for DED included three parameters: (1) the presence of subjective dry eye symptoms; (2) Schirmer’s I test value ≤ 5 mm at 5 min, or TBUT ≤ 5 s for the average value of three measurements; and (3) positive vital dye staining of the conjunctiva or cornea (LG or FL staining score of ≥3 out of 9). A positive finding for all the above parameters was considered grounds for a positive diagnosis of DED.

### 2.3. Evaluation Factors 

All participants underwent a general ophthalmic check-up for DED, including an assessment of LG conjunctival staining score, FL corneal staining score, TBUT, and Schirmer I test. Staining with LG and FL was performed by pipetting 2 µL each of lissamine and fluorescein sodium into the participant’s inferior lid margin, respectively. LG was used for corneal and conjunctival staining (graded as: 3 points each for nasal and temporal conjunctiva and cornea; range, 0–9) and FL for corneal staining (3 points each for upper, central, and lower cornea; range, 0–9). The average of three continuous measurements was used as the TBUT. Schirmer’s I test was performed without anesthesia by placing Schirmer’s test strip on the outer one-third of the lower temporal conjunctival fornix for 5 min (Fluores Ocular Examination Test Paper; Ayumi Pharmaceutical Co., Tokyo, Japan). All ocular surface staining procedures were performed by experienced ophthalmologists (M.M., Y.U., and M.U.). Ocular symptoms were evaluated using the DEQS developed in Japan. This questionnaire assesses six common eye symptoms (eye discomfort, dryness, pain, fatigue, eyelid heaviness, and hyperemia) and nine non-visual symptoms (difficulty in keeping eyes open because of symptoms; vision becomes blurry when engaging in activities that require sustained visual attention; light is too bright; eye symptoms worsen when reading newspapers, magazines, or books; eye symptoms reduce the ability to concentrate; eye symptoms interfere with work, housework, or studying; tend to avoid leaving the house because of eye symptoms; feel down due to eye symptoms). The questionnaire provides a summary scale (range, 0–100), where 100 is the worst possible outcome [38].

We analyzed the correlation between clinically evaluated factors and gal-3 concentration among the DE and non-DE groups, and among all participants, using the Spearman rank correlation test. Moreover, we divided participants into three groups based on the LG and FL scores (scores of 7–9, 4–6, and 0–3), and two groups based on TBUT, Schirmer’s I test value, and DEQS (≤5 and >5 s, ≤5 and >5 mm, and <32 and ≥32, respectively). 

### 2.4. Sample Collection

The tear fluid was collected from healthy controls and patients with DED to determine the gal-3 protein levels. First, we pipetted 50 µL of 0.9% sterile saline (Otsuka Pharmaceutical Factory Co., Osaka, Japan) into the unanesthetized inferior fornix and instructed participants to move their eyes to enable mixing of the tear fluid content [39]. Subsequently, tear fluid was collected by pipetting again. Individual tear samples were centrifuged for 30 min at 10,000× *g* at 4 °C, and all samples were promptly frozen at −80 °C until the next step. Protein concentration was determined using the Micro bicinchoninic acid Protein Assay Kit (Thermo Scientific, Waltham, MA, USA), according to the manufacturer’s protocol [37].

### 2.5. Analysis of Gal-3 Protein Concentration 

Tear samples containing 10–20 µg total protein were separated by electrophoresis on 10% sodium dodecyl sulfate-polyacrylamide gel electrophoresis gels and transferred onto nitrocellulose membranes (Bio-Rad Laboratories, Hercules, CA, USA). Then, membranes were blocked with 5% nonfat dry milk in Tris-buffered saline with Tween^®^ (TBST) for 2 h at room temperature, followed by incubation overnight at 4 °C with the following primary antibodies diluted in 5% nonfat dry milk in TBST: anti-galectin-3 (EP2775Y; 1:2500; Abcam, Cambridge, UK). Following incubation with the corresponding peroxidase-conjugated secondary antibody (1:2000; Thermo Scientific), positive binding was visualized using the SuperSignal^®^ West Substrate (Thermo Scientific) on a Sequi-Blot PVDF Membrane (Bio-Rad Laboratories). Band intensities were quantified by densitometry (ImageJ 1.52a, National Institutes of Health, Bethesda, MD, USA; in the public domain, available at http://imagej.nih.gov/ij/download/ accessed on 16 December 2021). Quantification of protein concentrations in the tear fluid was performed by immunoblotting as previously described [40]. The gal-3 concentration (ng/µg total protein) in tears was determined using a standard curve based on signal intensity generated with different concentrations of rhGal-3 and two internal controls of rhGal-3 per gel when analyzing the experimental samples. Recombinant human galectin-3 was expressed as previously reported [37].

### 2.6. Analysis of the Detection Rate of Gal-3C

During immunoblotting, we considered that the binding that was visualized and detected using ImageJ software was positive, whereas that not visualized or detected was negative. 

### 2.7. Statistical Analysis

Evaluation of significant differences between DE and non-DE, gal-3 protein concentration, and groups by severity of clinically evaluated factors was performed with Fisher’s exact, Mann–Whitney U, and Chi-square tests. The relationship between gal-3 protein concentration and the evaluated factors was analyzed using the Spearman rank correlation test. The significance value was set at *p* < 0.05. All statistical analyses were performed using Microsoft Office Excel 2016 software, version 1711 (Microsoft Corp., Redmond, WA, USA). 

## 3. Results

### 3.1. Clinical Evaluation of Dry Eye and Control Groups

In total, 42 patients were enrolled (80 eyes). The DED group included 52 eyes of 28 patients (2 men and 26 women), and the control group included 28 eyes of 14 healthy volunteers (3 men and 11 women). The volume of four tear samples in the dry eye (DE) group was too low to evaluate the gal-3 protein and total protein levels; thus, we excluded them from the analysis. The mean ages in these groups were 61.6 ± 14.7 (range, 33–83) and 61.0 ± 14.5 (range, 39–86) years, respectively. The etiology of DED included primary Sjögren’s syndrome (*n* = 19), secondary Sjögren’s syndrome (*n* = 3), graft versus host disease (*n* = 3), ocular cicatricial pemphigoid (*n* = 1), and no systemic ocular disease (*n* = 2) (Table 1).

The systemic disease background in the DE group was mainly primary Sjögren’s syndrome (*n* = 19) (Table 1). After analyzing the ophthalmic evaluation findings in the DE and non-DE groups, we found significant differences in the mean lissamine green (LG) corneal and conjunctival staining scores (DE: 5.67 ± 1.90; non-DE: 0 ± 0, *p* < 0.01), fluorescein (FL) corneal staining score (DE: 3.31 ± 2.41; non-DE: 0.14 ± 0.35, *p* < 0.01), tear film break-up time (TBUT) (DE: 2.64 ± 1.42; non-DE: 8.24 ± 1.97, *p* < 0.01), Schirmer’s I test value (DE: 2.02 ± 2.19; non-DE: 12.9 ± 10.37, *p* < 0.01), and dry eye questionnaire score (DEQS) (DE: 43.58 ± 22.17; non-DE: 6.42 ± 4.91, *p* < 0.01) (Table 2). 

### 3.2. Gal-3 Protein Concentration in the Tear Fluid of the Dry Eye (DE) and Non-DE Groups

To study the role of gal-3 in DED cases, we performed immunoblotting, as in previous studies, to quantify the levels of endogenous gal-3 in human tear samples. The standard curve of a two-fold serial dilution series of recombinant human galectin-3 (rhGal-3) was used to determine the linear range for gal-3 antibody detection. The protein bands were analyzed, and a linear response was found between 0.6 and 10 ng of rhGal-3 (Figure 1, left). 

We obtained the relative intensities of endogenous gal-3 in human tears, normalized to the amount of total protein, as previously described [37]. Gal-3 protein concentration significantly increased in the DE than in the control group (Mann–Whitney U test, *p* < 0.01). The mean concentrations of gal-3 in the DE and control groups were 0.88 ± 0.93 and 0.03 ± 0.07 ng/μg, respectively (Figure 1, right).

### 3.3. Gal-3 Protein Concentration and the Classification of DED Severity

We classified all participants into three groups based on their LG or FL staining score as follows: 0–3 points (28 eyes), 4–6 points (33 eyes) and 7–9 points (19 eyes). The results showed that gal-3 protein concentration in tears was significantly higher in the groups with higher LG and FL scores (*p* < 0.01). The mean concentrations of gal-3 in the groups with an LG of 7–9 (1.28 ± 1.02 ng/μg total protein) and 4–6 points (0.64 ± 0.78 ng/μg) were significantly higher than that in the group with an LG of 0–3 points (0.03 ± 0.07 ng/μg). Similarly, the participants were divided into three groups based on their FL staining score as follows: 0–3 (63 eyes), 4–6 (10 eyes), and 7–9 points (seven eyes). The mean gal-3 concentrations in the FL 7–9 (1.28 ± 1.02 ng/μg) and 4–6-point groups (0.64 ± 0.78 ng/μg) were significantly higher compared to that in the 0–3-point group (0.03 ± 0.07 ng/μg). Similarly, TBUT, Schirmer’s test value, and DEQS were also found to be significantly different between the two groups (TBUT ≤ 5 s (29 eyes) and > 5 s (51 eyes): 8.18 ± 8.71 and 1.61 ± 6.36 ng/μg, respectively (*p* < 0.01); Schirmer’s test value ≤ 5 mm (56 eyes) and > 5 mm (24 eyes): 7.94 ± 9.30 and 0.79 ± 2.32 ng/μg, respectively (*p* < 0.01); DEQS < 32 (23 eyes) and ≥ 32 (19 eyes): 2.56 ± 4.66 and 10.26 ± 8.38 ng/μg, respectively (*p* < 0.01)) (Figure 2). 

### 3.4. Correlation between Gal-3 Protein Concentration and the Severity of Clinically Evaluated Factors

To determine the association between gal-3 protein concentration and clinical parameters, we performed Spearman’s rank correlation test. We found that gal-3 protein concentration had a significant positive correlation with the LG staining score (R = 0.60; *p* < 0.01), FL staining score (R = 0.49, *p* < 0.01), and DEQS (R = 0.45, *p* < 0.01). Moreover, it had a negative correlation with TBUT (R = −0.40, *p* < 0.01) and Schirmer’s I value (R = −0.36, *p* < 0.01) (Figure 3).

### 3.5. Detection Rate of Cleavage of Gal-3 and the Severity of Clinically Evaluated Factors 

The detection rate of gal-3C in the DE group significantly increased compared to that in the control group (*p* < 0.01). The detection rate of gal-3C in the DE group was 19.2% (10 of 52), compared to 0% (0 of 28) in controls (Figure 4).

The detection rate of gal-3C was determined in each group separately. It was 26.3% (5/19), 20% (5/25), and 0% (0/36) in the LG 7–9-, 4–6-, and 0–3-point groups, respectively (*p* < 0.01). Among the FL groups, it was 57.1% (4/7), 20% (2/10), and 6.3% (4/63) in the 7–9-, 4–6-, and 0–3-point groups, respectively (*p* < 0.01). Among the TBUT groups, it was 17.6% (9/51) and 3.4% (1/29) in the ≤5 s and >5 s groups, respectively (*p* = 0.06). Similarly, it was 17.9% (10/56) when Schirmer’s I test value was ≤5 mm, and 0% (0/24) when it was >5 mm (*p* = 0.02). In the DEQS ≥ 33 and ≤32 groups, it was 21.6% (8/37) and 4.7% (2/43), respectively (*p* = 0.02). Among all the factors evaluated, we noted that a more severe DED resulted in a higher gal-3C detection rate (Figure 5).

### 3.6. Characteristics of Gal-3 Cleavage

The DE group was further divided into the gal-3C positive (*n* = 10) and negative groups (*n* = 42), and the clinically evaluated scores were compared between these two groups. There were statistically significant differences in the LG (positive: 6.8 ± 1.9; negative: 5.4 ± 1.78, *p* < 0.05) and FL scores (positive: 5.5 ± 2.9; negative: 2.8 ± 1.9, *p* < 0.01). However, the TBUT, Schirmer’s test results, DEQS, and gal-3 protein concentrations were not significantly different, though the patients in the gal-3C positive group were more likely to have DED of greater severity. The results were as follows: TBUT (positive: 2.1 ± 1.5; negative: 2.8 ± 1.4, *p* = 0.18), Schirmer’s test (positive: 1.2 ± 1.2; negative: 2.2 ± 2.3, *p* = 0.06), DEQS score (positive: 50.7 ± 25.2; negative: 43.0 ± 20.8, *p* = 0.36), and gal-3 protein concentration (positive: 12.7 ± 10.5; negative: 7.8 ± 8.8, *p* = 0.14) (Table 3).

## 4. Discussion

Gal-3 interacts with specific binding partners and is involved in several cellular activities, including apoptosis, cell migration, and angiogenesis [25,41,42,43]. Several studies have shown a high expression of gal-3 in breast [30], gastric [44], colorectal [45], and pancreatic cancers [46]. Therefore, gal-3 may be a diagnostic and/or prognostic marker. Another study showed that gal-3 was a possible biomarker for cardiac fibrosis and the severity of systemic inflammation in acute myocardial infarction cases [47]. Oikonomou et al. reported that gal-3 was a reliable evaluating marker of the preserved renal function in these patients with stable decompensated cirrhosis [48]. 

In ophthalmology, previous reports have demonstrated that the expression level of MUCs decreased on the ocular surface of patients with DED, while the formation of glycosylation was altered [36]. Moreover, the gal-3 concentration in the tears of such patients was reported to be higher than that in the controls, owing to diminished binding between MUCs and gal-3 [37]. Our data showed that there was a correlation between the severity of each ophthalmic examination finding and the gal-3 concentration in tears. To the best of our knowledge, this is the first report to reveal the association between the severity of DED and gal-3 concentration in tears. We suggested that gal-3 can be a biomarker for DED, and its measurement can be an alternative method for assessing the disruption of the glycocalyx barrier.

In this study, lissamine and fluorescein staining scores, which evaluate ocular surface disorders and the disruption of the ocular barrier, were highly correlated with gal-3 concentration in tears. LG staining has the same staining characteristics as that of rose Bengal staining [49,50,51], that is known to stain areas where mucin formation is impaired. Therefore, LG staining is a suitable alternative to rose Bengal staining because of its lesser intrinsic toxicity [52]. LG has also been reported to stain epithelial cells with damaged cell membranes, such as dead and degenerated cells [51,53]. Conjunctival disorders often precede corneal abnormalities in DED cases; therefore, LG staining might have resulted in a higher correlation than FL staining [54]. From the results of this analysis, the high correlation of LG staining with gal-3 concentration suggested that the former can reflect the disruption of the glycocalyx barrier.

DED is a condition where tear secretion decreases, and the disruption of the epithelial barrier is an essential diagnostic factor [55]. However, previous reports have shown that patients with reduced TBUT and reduced wettability on the ocular surface had the same subjective symptoms as those with disorders of the epithelial barrier [56,57]. Many of the subjective symptoms of DED are thought to be caused by a decrease in the wettability of tears on the ocular surface; therefore, DED treatment needs to improve the qualitative stability of tears. The diagnostic criteria for DED in Asia changed in 2017, and DED is diagnosed with only subjective symptoms and a shortage of TBUT. Although it is not mandatory to check the ocular surface damage and tear production by performing Schirmer’s test, they are important for the diagnosis of aqueous deficiency type DED and evaluation of the ocular damage [58]. The major causes of tear film instability are insufficient tear secretion and decreased epithelial wettability. The ocular surface glycocalyx, composed of MUCs and gal-3, is a key factor in maintaining the ocular surface barrier and wettability [59,60]. The decrease in wettability is thought to be caused by the disruption of the glycocalyx barrier formed by the binding of MUCs and gal-3 [15]; however, it is not easy to clinically evaluate the expression of MUCs in human corneal tissue. In this report, we found that there was a negative correlation between TBUT and gal-3 concentration in tears and a positive correlation between gal-3 and the extent of subjective dry eye symptoms as evaluated by the DEQS questionnaire. Therefore, the detection and quantification of gal-3 in tears is an effective alternative for assessing the expression of MUCs. Gal-3 can also be an effective marker for assessing reduced tear film instability and epithelial wettability, which could be important for dry eye evaluation in the future. Furthermore, we plan to develop the application for measurement of gal-3 in tears to estimate the severity of ocular surface glycocalyx barrier disruption.

Gal-3C is susceptible to rapid and efficient cleavage by matrix metalloproteinases (MMPs), especially MMP-2 and MMP-9 [61], which increase as DED becomes more severe. Moreover, gal-3C may be detected during the progression of breast [62] and prostate cancer [63], because of its role in chemotaxis, chemo-invasion, heterotypic aggregation, epithelial–endothelial cell interactions, and angiogenesis. In particular, the MMP-2 cleaved gal-3 protein is known to be correlated with angiogenesis in tumor progression in the early stage [64], leading Nangia-Makker et al. to suggest that gal-3C can be a new therapeutic target [41]. However, in the field of ophthalmology, the function of gal-3C remains obscure. Nevertheless, we showed that the detection rate of gal-3C in the tear samples of patients with DED increased with the severity of dry eye. Gal-3C may play an essential role in the deterioration of DED and could be a novel biomarker for evaluating the severity of ocular surface barrier disruption and inflammation. 

A limitation of this study was the small sample size to perform accurate statistical analysis. Additionally, there was a difference in sample size among the three groups divided based on their FL staining score as follows: 0–3 (63 eyes), 4–6 (10 eyes), and 7–9 (seven eyes)-point groups. The relatively small sample size was attributed to the difficulty in collecting tear sample from patients. As we obtained encouraging results in this study, we are planning to conduct a large-scale study prior to implementing our results to the clinic. As a second limitation, we did not check the positivity of Meibomian gland dysfunction (MGD) in the participants. MGD may influence the pathology of DED; however, we could not reveal the relationship between gal-3 and MGD.

In summary, we found that Gal-3 in the tears of patients with DED was significantly correlated with ophthalmic evaluation factors, especially the LG staining score, which is used to evaluate the disruption of ocular surface glycocalyx. Gal-3C was also found to have a higher detection rate in more severe cases of DED. These data suggested that gal-3 and gal-3C may have the possibility of being novel ocular surface biomarkers to evaluate the severity of ocular surface barrier disruption.

## Figures and Tables

**Figure 1 jcm-11-00066-f001:**
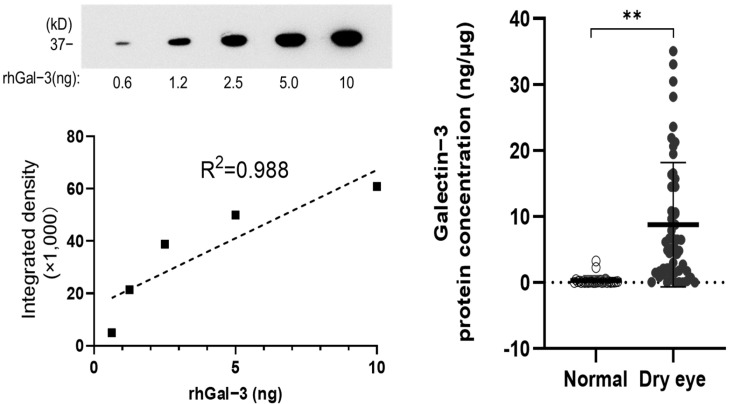
(**Left**) Standard curve of a two-fold serial dilution series of recombinant human galectin-3 (rhGal-3). (**Right**) Analysis of galectin-3 concentrations in tears included 52 dry eye disease samples (28 patients), and 28 normal eye samples (14 healthy volunteers) by immunoblotting. ** *p* < 0.01.

**Figure 2 jcm-11-00066-f002:**
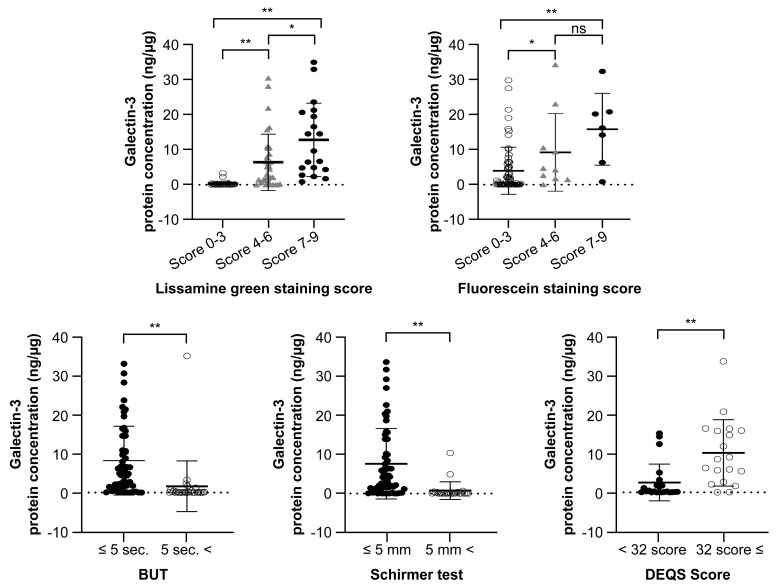
Based on the LG score, there were significant difference between the 7–9 and 0–3 (*p* < 0.01), 7–9 and 4–6 (*p* < 0.05), and 4–6 and 0–3-point groups (*p* < 0.01). FL score was also significantly different between the 7–9 and 0–3 (*p* < 0.01), 7–9 and 4–6 (*p* < 0.05), and 4–6 and 0–3-point groups (*p* < 0.01). Similarly, TBUT, Schirmer’s value, and DEQS were different between the two groups (TBUT ≤ 5 and >5 s, *p* < 0.01; Schirmer’s test ≤ 5 and >5 mm, *p* < 0.01; DEQS score < 32 and ≥32, *p* < 0.01. ** *p* < 0.01; * *p* < 0.05; n.s., not significant. LG, lissamine green staining; FL, fluorescein staining; DEQS, dry eye questionnaire score; TBUT, tear film break-up time.

**Figure 3 jcm-11-00066-f003:**
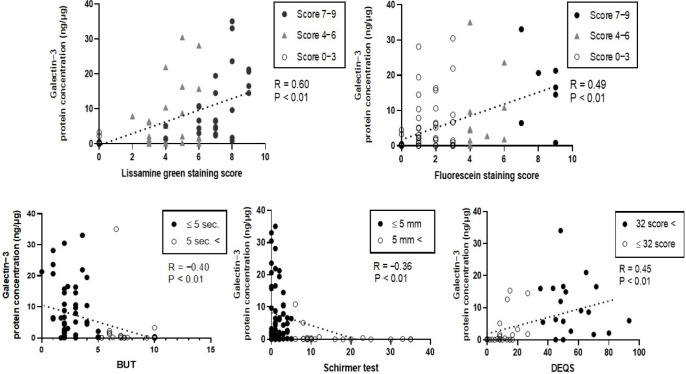
Correlation between gal-3 protein concentration and ophthalmic evaluation factors. Significant positive correlations were found between gal-3 concentration and LG score (R = 0.60; *p* < 0.01), FL score (R = 0.49, *p* < 0.01), and DEQS (R = 0.45, *p* < 0.01). Negative correlations were found between gal-3 concentration and TBUT (R = −0.40, *p* < 0.01), and Schirmer’s I value (R = −036, *p* < 0.01). LG, lissamine green staining; FL, fluorescein staining; DEQS, dry eye questionnaire score; TBUT, tear film break-up time.

**Figure 4 jcm-11-00066-f004:**
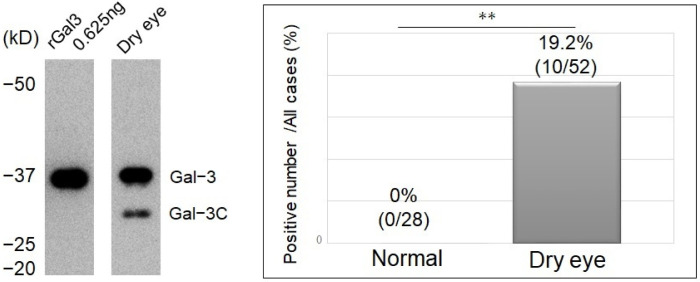
The detection rate of gal-3C in patients and controls. Gal-3C was detected in 19.2% of patients with dry eye disease, and not detected in controls (*p* < 0.01) ** *p* < 0.01; gal-3C, cleaved galectin-3.

**Figure 5 jcm-11-00066-f005:**
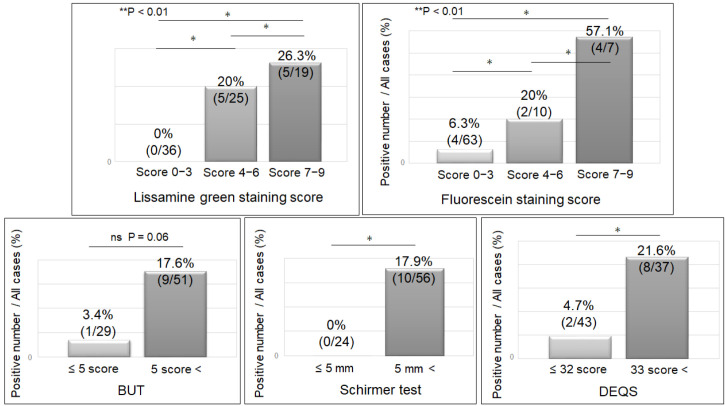
The detection rate of gal-3C in tears of patients versus the severity of dry eye disease. LG (*p* < 0.01), FL (*p* < 0.01), Schirmer’s test (*p* < 0.05), and DEQS (*p* < 0.05) were significantly different between the groups. The detection rate of gal-3C increased with the severity of dry eye disease. * *p* < 0.05, ** *p* < 0.01; n.s., not significant. LG, lissamine green staining; FL, fluorescein staining; DEQS, dry eye questionnaire score; TBUT, tear film break-up time; gal-3C, cleaved galectin-3.

**Table 1 jcm-11-00066-t001:** Background of systemic disease in the dry eye disease group.

	*n*
Primary Sjögren’s syndrome	19
Secondary Sjögren’s syndrome	3
Graft versus host disease	3
Ocular cicatricial pemphigoid	1
No systemic disease	2
Total	28

**Table 2 jcm-11-00066-t002:** The average values of ophthalmic evaluation factors in patients and controls.

Ophthalmic Evaluation Factors	Dry Eye Disease	Control	*p*
LG staining score (0–9 points)	5.67 ± 1.90	0	<0.01
FL staining score (0–9 points)	3.31 ± 2.41	0.14 ± 0.35	<0.01
TBUT (seconds)	2.64 ± 1.42	8.24 ± 1.97	<0.01
Schirmer test (mm)	2.02 ± 2.19	12.9 ± 10.37	<0.01
DEQS	43.58 ± 22.17	6.42 ± 4.91	<0.01

LG, lissamine green; FL, fluorescein; TBUT, tear film break-up time; DEQS, dry eye questionnaire score.

**Table 3 jcm-11-00066-t003:** The analysis of the ophthalmic evaluation factors in the Gal-3C positive or negative groups.

Gal-3C	Positive	Negative	*p*
LG staining score (0–9 points)	6.8 ± 1.9	5.4 ± 1.78	<0.05
FL staining score (0–9 points)	5.5 ± 2.9	2.8 ± 1.9	<0.01
TBUT (seconds)	2.1 ± 1.5	2.8 ± 1.4	0.18
Schirmer test (mm)	1.2 ± 1.2	2.2 ± 2.3	0.06
DEQS	50.7 ± 25.2	43.0 ± 20.8	0.36
Gal-3 protein concentration	12.7 ± 10.5	7.8 ± 8.8	0.14

## Data Availability

The data that support the findings of this study are available within the article and supplemental data or from the corresponding author upon reasonable request.

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
