# Peer review of "Analysis of the Association between Galectin-3 Concentration in Tears and the Severity of Dry Eye Disease: A Case-Control Study"

_jcm, 2021, doi:10.3390/jcm11010066_

Round 1

Reviewer 1 Report

This is a very interesting study that aims to investigate the relationship between the severity of dry eye disease and galectin-3 concentration and its cleavage in tear fluid. It is preceded of a previous study that revealed that gal-3 and gal-3C could be detected better in tears of patients with dry eye disease compared to normal individuals.

Introduction gives enough information to understand the importance of galectin-3 in tears stability. It is also well referenced and with the aim of the study described.

Methodology is also fine, but I have some doubts and minor recommendations to improve the manuscript:

  • I would include the date of the study (when patients were enrolled)
  • It is important to clarify why authors have 80 eyes from 42 patients. It has been well explained in results, so perhaps lines 82-89 should be part of the results (it is a descriptive analysis of the sample) and in methodology authors should include how the enrolled the sample and the inclusion and exclusion criteria
  • I would like to know if the subjects were using some treatment that could influence the results or not.
  • I also miss information about the ethical committee and the informed consent of subjects.

Results are well explained, although I think that the symbol < should not be included in the p values that are over 0.001 (lines 264-271 and table 3)

Regarding to discussion, authors discuss the results including all the limitations of the study.

Author Response

Comment and Suggestions

Introduction gives enough information to understand the importance of galectin-3 in tears stability. It is also well referenced and with the aim of the study described. Methodology is also fine, but I have some doubts and minor recommendations to improve the manuscript.

Response

The authors would like to thank the reviewer for his/her constructive critique to improve the manuscript. We have made every effort to address the issues raised and to respond to all comments. Please, find next a detailed, point-by-point response to the reviewer's comments. We hope that our revisions would meet the reviewer’s expectations.

Comment #1.

I would include the date of the study (when patients were enrolled)

Response #1.

Thank you very much for reviewer’s suggestion. We add the date of time for enrolled patients. (new line 34-35)

Comment #2.

It is important to clarify why authors have 80 eyes from 42 patients. It has been well explained in results, so perhaps lines 82-89 should be part of the results (it is a descriptive analysis of the sample) and in methodology authors should include how the enrolled the sample and the inclusion and exclusion criteria

Response #2.

We perfectly agree with the reviewer’s suggestion. We change the position of the original sentence 82-89 lines from the methods section to the results sections (new line 172-176). Furthermore, we set the inclusion and exclusion criteria in methods section (new line 172-176).

Comment #3.

I would like to know if the subjects were using some treatment that could influence the results or not. I also miss information about the ethical committee and the informed consent of subjects.

Response #3.

This indication point is very important, but we don’t know which treatment could have an impact to the results or not. In the next study, we should make a study design to reveal this point.

We described information about the ethical committee and the informed consent of subjects in Institutional Review Board Statement section (Line 386-389).

Comment #4

Results are well explained, although I think that the symbol < should not be included in the p values that are over 0.001 (lines 264-271 and table 3).

Regarding to discussion, authors discuss the results including all the limitations of the study.

Response #4

We appreciate the reviewer’s comments. We checked all the symbol “ < “ in results, and changed the notation adequately.

Reviewer 2 Report

The Authors present a report revealing associations between the severity of dry eye disease and gal-3/gal-3C concentration in tears, and their continued efforts on such an important topic should be lauded.

In the Discussion section (lines 336-337) more information about the supposed function of these galectins in the eye should be acknowledged and reported.

Reference 48 (lines 282-283) should be revised, as the study tried to clarify the role of gal-3 in patients with stable decompensated cirrhosis, evaluating its association with the preserved renal function of these patients.

A reference should be added regarding the role of Gal-3 in pancreatic cancer (line 279).

Author Response

Comment and Suggestions

The Authors present a report revealing associations between the severity of dry eye disease and gal-3/gal-3C concentration in tears, and their continued efforts on such an important topic should be lauded.

Response

The authors would like to thank the reviewer for his/her constructive critique to improve the manuscript. We have made every effort to address the issues raised and to respond to all comments. Please, find next a detailed, point-by-point response to the reviewer's comments. We hope that our revisions would meet the reviewer’s expectations.

Comment #1

In the Discussion section (lines 336-337) more information about the supposed function of these galectins in the eye should be acknowledged and reported.

Response #1

We appreciate the reviewer’s comment, and improved the information about the supposed function of these galectins on ocular surface. (new line 346-349)

Comment #2

Reference 48 (lines 282-283) should be revised, as the study tried to clarify the role of gal-3 in patients with stable decompensated cirrhosis, evaluating its association with the preserved renal function of these patients.

Response #2

We revised the information of reference #48 as reviewer’s comments (new line 304-306).

Comment #3

A reference should be added regarding the role of Gal-3 in pancreatic cancer (line 279).

Response #3

We added regarding the role of Gal-3 in pancreatic cancer (reference #46) (new line 302).